# Lipid Profile in Children Born Small for Gestational Age

**DOI:** 10.3390/nu15224781

**Published:** 2023-11-15

**Authors:** Justyna Zamojska, Katarzyna Niewiadomska-Jarosik, Beata Kierzkowska, Marta Gruca, Agnieszka Wosiak, Elżbieta Smolewska

**Affiliations:** 1Department of Pediatric Cardiology and Rheumatology, Medical University of Lodz, 90-419 Lodz, Poland; kasiajarosik@wp.pl (K.N.-J.); beata.kierzkowska@umed.lodz.pl (B.K.); marta.gruca@onet.eu (M.G.); elzbieta.smolewska@umed.lodz.pl (E.S.); 2Institute of Information Technology, Lodz University of Technology, 90-924 Lodz, Poland; agnieszka.wosiak@p.lodz.pl

**Keywords:** children, lipid profile, cholesterol, small for gestational age

## Abstract

Background: Lipid disorders are one of the risk factors for cardiovascular diseases. The aim of the study was to estimate the lipid profile in early childhood in the population of Polish children born small for gestational age (SGA). Materials and Methods: The study included 140 patients (93 SGA children and 47 controls) aged 5 to 11 years. All the subjects underwent a physical examination and blood laboratory tests for the glucose and lipid profiles. The SGA group was divided into subgroups, i.e., symmetrical and asymmetrical intrauterine growth restriction (IUGR). Results: Blood sample analysis revealed higher levels of total cholesterol (SGA group 190.61 ± 24.66 mg/dL vs. controls 143.23 ± 23.90; *p* < 0.001). The analysis of particular cholesterol fractions showed significantly higher mean values of triglycerides and LDL cholesterol as well as lower mean values of HDL cholesterol in SGA children. Children in both groups did not differ significantly in terms of weight or body mass index. A statistically significantly higher glucose concentration was observed in SGA patients with the symmetrical type of IUGR. Analyzing the differences regarding metabolic factors, we obtained a statistically significant difference only in fasting glucose concentration (asymmetrical IUGR = 90.56 ± 10.21 vs. symmetrical IUGR = 98.95 ± 14.79; *p* < 0.001). Conclusions: Children born SGA, even those not suffering from overweight or obesity in their early childhood, have an abnormal lipid profile, which may contribute to the development of cardiovascular diseases in adulthood.

## 1. Introduction

In the 1990s, the concept of fetal programming emerged as having a risk of developing type 2 diabetes, hypertension, and dyslipidemia in patients born small for gestational age (SGA) [1]. In the same year, Barker put forward a hypothesis suggesting that poor intrauterine growth caused by fetal malnutrition was associated with an increased incidence of cardiovascular diseases (CVDs) in adulthood. The hypothesis posited that fetuses receiving insufficient nutrition are more prone to develop abnormally high body weight. It is widely known that overweight and obesity contribute to the onset of cardiovascular and endocrine diseases (e.g., diabetes) in these individuals [2]. Moreover, the fetal origin hypothesis has expanded the conventional focus on adult health behaviors, such as smoking, appropriate levels of physical activity, and type of nutrition, to previously mentioned habitual factors that can influence prenatal welfare. Barker’s hypothesis has inspired scientists to study the phenomenon of fetal programming in a more profound way. Birth weight is strongly correlated with the gender of the newborn, maternal and paternal anthropometric parameters, gestational age, and maternal comorbidities, i.e., diabetes, hypertension, or smoking [3].

The issue of low birth weight (LBW) remains a significant public health concern worldwide. Birth weight is regarded as one of the predictors of child mortality and morbidity. LBW newborns, as defined by the World Health Organization (WHO), are those weighing less than 2500 g (up to and including 2499 g) [4]. The term LBW refers to an absolute weight of <2500 g, independently of gestational age.

SGA is defined as birth weight below the tenth centile for gestational age and affects about 5 to 10% of neonates in developed countries [5]. The study revealed that 52% of unconfirmed stillbirths are caused by SGA, which also accounts for 10% of perinatal mortality. Furthermore, it is noteworthy that a significant proportion of unaccounted-for fetal deaths, namely, 72%, are attributed to low birth weight [6]. The “thrifty phenotype” hypothesis proposes that metabolic adaptations in early life support the survival of an organism by determining a proper growth trajectory in response to unfavorable intrauterine environmental simulation [7]. It is estimated that the global incidence of low birth weight ranges from 15% to 20% of all births, corresponding to more than 20 million births annually [8].

Children born small for gestational age can present two types of intrauterine growth restriction—symmetrical and asymmetrical. The conditions develop during organogenesis in the first or second trimester of pregnancy. The first type accounts for approximately 20–25% of all cases. In this type, all fetal dimensions are decreased in terms of anthropometrical dimensions as well as internal organs, which is usually accompanied by a permanent reduction in growth potential. The second type (asymmetrical IUGR) accounts for most cases, i.e., about 75–80%. This type is formed at the end of the second and third trimesters of pregnancy due to abnormal cell growth, not their amount. In the asymmetrical type, neonates are born with low weight, while other anthropometric parameters, such as body length, head circumference, and chest circumference, remain normal. 

For this reason, the ponderal index (calculated based on the formula birth weight (g) × 100/length (cm^3^)) is lower than in the symmetrical type [9,10,11]. According to the nomenclature, the two terms, i.e., intrauterine growth restriction and fetal growth restriction, are used interchangeably. They both refer to the prenatal diagnosis of growth restriction. Moreover, both these terms (IUGR and fetal growth restriction) are often used conversely with small for gestational age (SGA) [12]. A number of infants with SGA have IUGR, and indeed many infants with IUGR also have SGA. However, SGA cannot be used as a marker for IUGR because some infants with IUGR will have a birth weight greater than the 10th percentile [11]. Therefore, when distinguishing between SGA and IUGR, it is important to use fetal growth curves customized to constitutional factors to separate normal SGA infants from those with IUGR. It is important to highlight the differences between LBW, SGA, and IUGR. The term LBW is a separate nomenclature and should not be equivalently used with IUGR/SGA. This is because the definition of LBW is based on birth weight (less than 2500 g) regardless of gestational age, sex, race, and clinical features [11]. The definition of SGA refers to newborns whose birth weight is below the 10th percentile for a given gestational age or two standard deviations below population norms on growth charts. It should be mentioned that the definition of SGA only considers birth weight without taking into account intrauterine growth and physical characteristics at birth. Moreover, it must be added that we distinguish SGA and preterm SGA depending on the duration of pregnancy. The definition of IUGR is a typically clinical concept and applies to infants born with features of malnutrition and intrauterine growth retardation, independent of birth weight percentile [11]. To assess fetal growth, Roth’s ponderal index (PI), a ratio of body weight to length, was used. In newborns with IUGR, where the accumulation of adipose tissue and muscle mass are reduced, the ponderal index is lower [13].

LBW is associated with many long-term complications. Numerous epidemiological studies show that low birth weight, regardless of other risk factors that appear later in life, can lead to cardiovascular diseases (especially hypertension and hyperlipidemia) and stroke, as well as increased mortality in adulthood [14,15]. Infants born with LBW may suffer from persistent neurological impairments, language development disorders, diminished academic performance, and a higher likelihood of chronic illnesses, such as the aforementioned cardiovascular diseases and diabetes [16]. In general, the occurrence of LBW can be triggered by fetal, placental, and maternal factors. Despite their different etiologies, these causes often share a common final pathway of inadequate uteroplacental perfusion and fetal nutrition [17]. Most children with SGA compensate for limited intrauterine growth by catching up early [18].

Catch-up growth (especially in terms of body weight) has been shown to increase the occurrence of some cardiometabolic risk factors, including resistance to insulin, overweight, and obesity, already in childhood [19]. The metabolic syndrome is a cluster of cardiovascular risk factors, including obesity, alterations in glucose-insulin metabolism, hypertension, and dyslipidemia [18]. The effect of birth weight on lipid levels in childhood is somewhat controversial. Low birth weight appears to be associated with unfavorable lipid levels [20]. It is considered that poor catch-up growth in height may increase the risk of high total cholesterol [21].

Studies have also revealed a positive correlation between elevated levels of total cholesterol, LDL cholesterol, and apolipoprotein B in adults born SGA and a small abdominal circumference observed in newborns after birth [22]. There are relatively few reports referring to research on lipids and lipoproteins in the blood serum of SGA children [21,23,24,25,26,27,28,29]. Although their results vary depending on the studied population, it is worth emphasizing that assessing of the lipid profile in SGA children is recommended in certain regions [30].

The long-term effects of low birth weight have been analyzed in many publications [2,16,18,19,31,32,33,34]. As we mentioned earlier, low birth weight in the long perspective is most related to cardiovascular and metabolic diseases in adulthood [2]. It has been shown that adults born SGA had higher systolic and diastolic blood pressure compared to adults born AGA [31]. According to an article published in the European Journal of Endocrinology, SGA adults had higher fat mass with central adiposity, insulin resistance, and an improper lipid profile [32]. In addition, it was found that carotid intima-media thickness (cIMT) was higher in SGA-born adults with spontaneous catch-up growth compared to adults born with AGA [31]. Regarding the catch-up growth phenomenon, it was also revealed that in SGA girls, during growth and development, adipose deposition and insulin resistance can develop, leading to metabolic syndrome in adulthood [33]. It is also shown that young adults born with SGA are more frequently obese and have lower serum leptin levels compared to AGA controls [34].

Cardiovascular disease prevention is a comprehensive set of measures undertaken at the population level or targeted at individuals to eliminate or minimize the adverse effects of cardiovascular disease and its associated impairments [35]. The preventive efficacy of such measures has been demonstrated in numerous population-based studies, including the CARDIA study [36].

Despite many studies on SGA patients currently [5,6,18,21,23,24,25,26,27,28,29,30], the increased exposure of children to risk factors for the development of CVD sheds new light on this patient group. A sedentary lifestyle, abnormal dietary habits, and their consequent lipid metabolism disorders contribute to the development of cardiovascular disease in the pediatric population [37]. Patients with SGA are at higher risk of developing CVD than their healthy peers [18]. Therefore, it is important to conduct research in this group of patients in order to detect any metabolic disorders that may develop early.

This study is a retrospective work to determine whether children from the Polish population born small for gestational age are affected by abnormalities in the lipid profile in their early years.

## 2. Materials and Methods

The study included 93 children (50 girls, 43 boys) aged 5–11 years, randomly selected from an outpatient clinic at the Pediatric Cardiology and Rheumatology Department of the Medical University of Lodz, born on time and small for gestational age, i.e., with birth weight below the tenth centile according to gestational age (SGA—small for gestational age), assessed prenatally based on fetal ultrasound parameters. The control group comprised 47 healthy children (24 girls, 23 boys), born with normal birth weight (appropriate for gestational age—AGA), age- and sex-matched to the analyzed group. Gestational age was calculated based on the date of the mother’s last menstrual period.

The small for gestational age subjects were divided into two subgroups, i.e., symmetrical (*N* = 43) and asymmetrical IUGR (*N* = 50). The division into subgroups was made based on data from the prenatal history—according to ultrasound measurements of the head, abdominal, and chest circumference; biparietal diameter; and femur length. Then, it was based on the analysis of anthropometric measurements at birth and the ponderal index. The following formula was used to calculate the ponderal index: birth weight (g) × 100/length (cm^3^). Symmetrical IUGR had all parameters reduced and a normal ponderal index, i.e., more than 2. Asymmetrical IUGR had reduced weight and length, normal head circumference, and a ponderal index less than 2.

Mothers who had multiple pregnancies; high blood pressure during pregnancy; diabetes before pregnancy; liver problems during pregnancy; infection with syphilis or HIV; or systemic diseases like hypertension, diabetes, rheumatism, thyroid disease, chronic kidney disease, and obesity were not qualified for our study. Furthermore, the exclusion criteria in the study were illicit drug use during pregnancy, maternal smoking, or alcohol consumption. Evidence for chromosomal or infectious etiology of SGA, hypothyroidism, and systemic or acute disease (e.g., kidney or liver dysfunction, hypertension, gastrointestinal, nephrological, neurological, or cardiac diseases) were also exclusion criteria for pediatric participants. Children who had a history of taking antibiotics or other symptomatic drugs used for infections were not excluded. Only pediatric patients receiving other pharmacotherapies were not included. Informed consent was obtained from all the children’s parents. All the subjects underwent a physical examination, during which demographic and anthropometric data were collected. The data included information about birth weight and gestational age. Current weight, height, and body mass index (BMI) were measured. When measuring body mass using a calibrated scale, each participant was dressed in light clothing and barefoot. For the calculation of BMI, the formula weight (kg)/height (m^2^) was applied. Blood samples were collected from all the patients for fasting glucose and the lipid profile, including individual lipoprotein fractions, i.e., overall cholesterol, high-density lipoprotein cholesterol (HDL-cholesterol), low-density lipoprotein cholesterol (LDL-cholesterol), and triglycerides.

The blood samples were collected in the morning from the children who had fasted for at least 12 h.

The distribution of parameters was tested for normality using the Shapiro–Wilk W test. Differences in characteristics between the control group and patients with SGA, as well as between symmetrical and asymmetrical SGA cases, were assessed using the Mann–Whitney U test. Relationships were evaluated using Spearman’s correlation coefficient. Scatterplots were drawn to illustrate the correlations between fasting glucose levels and birth weight as well as actual body weight. Results were considered statistically significant if the *p*-value was less than 0.05.

The study was approved by the Medical Ethical Committee of the Faculty of Health Sciences, Lodz University (No. RNN/150/09/KB).

## 3. Results

All the children were examined at the Pediatric Cardiology and Rheumatology Department of the Medical University of Lodz. At the time of the study, they were in good condition, without any chronic illnesses or drug-taking history.

The medical history investigation showed a statistically significant distinction regarding birth weight (SGA group—2545.22 ± 209.07 g vs. control group—3370.29 ± 503.11 g; *p* < 0.001). However, there was no significant difference between the groups regarding gestational age. The physical examination revealed no statistically significant differences between the mean values of body weight and body mass index (BMI), while the SGA group subjects were significantly smaller than the control peers (Table 1).

We divided the children with SGA into the symmetrical (*N* = 43) and asymmetrical (*N* = 50) types of intrauterine growth restriction (Table 2). We found a significant difference in the ponderal index value between these groups (mean symmetrical 2.02 ± 0.23 g/cm^3^ vs. mean asymmetrical 1.75 ± 0.16 g/cm^3^; *p* < 0.00001). In the symmetrical SGA group, the current height (121.51 ± 9.34 vs. 125.73 ± 10.34; *p* < 0.05) and body weight (23.22 ± 7.59 vs. 26.14 ± 9.13; *p* < 0.05) were statistically significantly lower than in the group of children with the asymmetrical SGA.

The analysis of blood samples revealed statistically significantly higher fasting glucose levels (*p* < 0.001) in the SGA group. Similar findings were observed for the mean total cholesterol levels (SGA group 190.61 ± 24.66 mg/dL vs. control subjects 143.23 ± 23.90; *p* < 0.001). The analysis of particular cholesterol fractions showed significantly higher mean values of triglycerides, LDL cholesterol, and HDL cholesterol in the children born SGA (Table 3).

Analyzing the differences regarding metabolic factors between the subgroups of children with asymmetrical and symmetrical IUGR (Table 4), we obtained a statistically significant difference only in fasting glucose concentration (mean glucose level for asymmetrical IUGR = 90.56 ± 10.21 vs. symmetrical IUGR = 98.95 ± 14.79; *p* < 0.001). Total cholesterol, HDL cholesterol, LDL cholesterol, and triglycerides were not statistically significantly different in the IUGR subgroups. However, when analyzing the SGA group according to sex, our study showed statistically significant higher HDL cholesterol levels in males compared to females (SGA males = 75.55 ± 12.80 vs. SGA females = 67.91 ± 12.91; *p* < 0.05). Comparably, we also obtained significantly higher HDL cholesterol values in the group of boys, taking into consideration the division into symmetric and asymmetric hypotrophy. However, we achieved statistical significance only in the group of boys with asymmetric hypotrophy (77.38 ± 14.30 vs. 67.87 ± 12.07, *p* < 0.05). As for the remaining metabolic parameters, the values of total cholesterol and LDL cholesterol were higher in the SGA group in boys, and triglycerides were higher in girls, but without statistical significance.

We studied numerous correlations in both groups of patients. Surprisingly, few correlations reached statistical significance (see Table 5 and Table 6).

In the SGA group children, there was a negative correlation between birth weight and fasting glucose levels (r = −0.17; *p* = 0.033), total cholesterol levels (r = −0.01; *p* = 0.921), LDL cholesterol levels (r = −0.02; *p* = 0.848), and triglyceride levels (r = −0.04; *p* = 0.734), and there was a positive correlation between birth weight and HDL cholesterol levels (r = 0.09; *p* = 0.415). None of these correlations were statistically significant. In the AGA group, the correlation results were slightly different. There were positive correlations between birth weight and fasting glucose levels (r = 0.33; *p* = 0.054), total cholesterol levels (r = 0.03; *p* = 0.861), HDL cholesterol levels (r = 0.16; *p* = 0.352), and triglyceride levels (r = 0.02; *p* = 0.894), and there were negative correlations between birth weight and LDL cholesterol levels (r = −0.01; *p* = 0.946). In this group, the correlations did not reach the level of statistical significance either.

The statistically significant negative correlations in the SGA group of children were found between the fasting glucose level and birth height (r = −0.40; *p* < 0.001) (Figure 1), as well as between the fasting glucose level and current body weight (r = −0.22; *p* = 0.034) (Figure 2).

In the AGA group, we found this correlation to be positive, however, with no statistical significance (r = 0.18; *p* = 0.305). Contrary to our expectations, we found a negative correlation between total cholesterol and actual body weight as well as BMI in the SGA group, but it was not statistically significant (r = −0.01; *p* = 0.922 and r = −0.09; *p* = 0.393, respectively). In the AGA group, these correlations were different. We observed a positive, insignificant correlation between total cholesterol levels and the patient’s actual weight (r = 0.05; *p* = 0.785) and BMI (r = 0.04; *p* = 0.839).

In the SGA group, current body weight also correlated negatively with LDL cholesterol (r = −0.03; *p* = 0.811) and HDL cholesterol levels (r = −0.002; *p* = 0.979), whereas with triglyceride levels, the correlation was positive (r = 0.08; *p* = 0.462). These correlations were not statistically significant. 

In terms of BMI in the SGA group, and for fasting glucose level (r = −0.16; *p* = 0.118), LDL cholesterol (r = −0.08; *p* = 0.448), and HDL cholesterol levels (r = −0.06; *p* = 0.545), we found statistically insignificant negative correlations, whereas a positive correlation was observed for triglycerides (r = 0.10; *p* = 0.320). In the AGA group, we found a negative correlation with BMI for HDL cholesterol (r = −0.381; *p* = 0.023) and, unlike in the SGA group, a negative correlation for triglycerides (r = −0.17; *p* = 0.329). In the AGA group, a positive correlation was revealed between BMI and fasting glucose (r = 0.10; *p* = 0.577) as well as LDL cholesterol (0.06; *p* = 0.746). All the correlations mentioned above were not statistically significant. 

Regarding metabolic parameters and the subgroup of SGA patients (Table 7 and Table 8), we found a statistically significant negative correlation in the patients with the asymmetrical type of IUGR between birth length and fasting glucose concentration (r = −0.27; *p* = 0.05). For both types of IUGR, we observed a downward correlation between birth weight and current body weight (r = −0.06; *p* = 0.66 and r = −0.008; *p* = 0.59)—however, this was not statistically significant. Also, the BMI of the patients correlated positively with the glucose level in the subjects with asymmetrical IUGR and negatively in the subjects with the symmetrical type (r = 0.03; *p* = 0.84 and r = −0.15; *p* = 0.33). These correlations were not statistically significant. As regards total cholesterol levels and concentrations of HDL cholesterol, LDL cholesterol, and triglycerides, we found statistically significant positive correlations between birth length as well as total cholesterol levels and the HDL cholesterol level (r = 0.45; *p* = 0.003) in asymmetrical IUGR patients. There were no other statistically significant correlations between the lipid profile and BMI, birth, and actual weight in the two IUGR subgroups.

## 4. Discussion

Our results show significant differences in the lipid profile between the SGA group and the AGA controls. As it is known from the literature, prenatal development can affect CVD outcomes. Factors influencing prenatal and postnatal growth may affect future lipid profiles and their clinical manifestations in adulthood [38]. 

Since the 1990s, several controversial studies have supported the hypothesis that low birth weight can lead to metabolic changes in these individuals in later life.

In their observational studies, Howe et al. found a correlation between postnatal growth and the lipid profile [39]. The Avon Longitudinal Study of Study of Parents and Children, ALSPAC, conducted since 1991, has revealed a positive correlation between the ponderal index in infancy and levels of triglycerides and LDL cholesterol in patients at 15 years of age. These authors also revealed a negative correlation between this index and HDL cholesterol levels in the same group. Other researchers, in their prospective study (Amsterdam Born Children and their Development), also showed a negative correlation for HDL cholesterol levels and a positive correlation for triglycerides and postnatal weight gain in children aged 5–6 years. The postnatal weight was observed to change over one to three months [40]. In our study, we obtained similar results regarding HDL cholesterol but opposite results regarding triglycerides. This may be due to the older age of the children we examined or a different type of diet also due to age. Similar to other researchers, our analysis showed significantly higher mean values of fasting glucose levels in the children born SGA [29].

Yu et al., whose study only concerned 8-year-old girls, showed, like us, a statistically significant lower height and body weight in SGA patients. Contrary to our results, BMI was significantly lower. This difference may result from examining only the female population [33]. Our results are also consistent with the ponderal index. We found a significantly lower value of the ponderal index in the patients with the asymmetrical type of IUGR [9,10,11]. 

Nevertheless, some previous works are not directly comparable with our paper because of differences in age, weight, and height of the examined children.

Our results correspond to reports of Donker et al., who found a relationship between higher triglyceride levels and low birth weight in children in a similar age group (7–11 years) [23]. Prevalence rates in the upper decile of serum lipid concentrations for children with a low birth weight (±2500 g) compared to children with a birth weight > 2500 g were calculated for each race–gender group. Based on a study by American researchers, it was shown that white boys with a low birth weight were, in a higher percentage than expected, in the highest range of triglyceride concentrations (0.01 ± *p* ± 0.05) [23]. This study was the first to demonstrate an association between low birth weight of children and their high triglyceride levels in the future. In our study, we did not observe statistically significant differences according to triglyceride levels between boys and girls, but interestingly, we obtained higher values of HDL cholesterol in male SGA patients. This difference was also statistically significant in male group with asymmetrical SGA. For patients with symmetrical SGA, this value was without statistical meaning. 

Antal et al., on the other hand, when analyzing children aged 14–16 years, found no differences in lipid concentrations between individuals with insufficient birth weight and those with normal birth weight. It should be mentioned that the examined children differ according to fetal age, including those born prematurely [24]. Findings similar to ours were presented by Koklu et al. in a group of infants with SGA of different ages than our subjects. The authors observed notably elevated levels of triglycerides compared to the AGA group. In our analysis, this difference was also noticeable. They also assessed aortic intima-media thickness, showing higher values for SGA patients and a strong correlation with triglyceride levels [41]. We did not assess cIMT in our study, but the association between carotid artery IMT complex thickness and lipid disorders has been increasingly reported in scientific studies [42]. 

Spanish researchers studied 135 newborns with intrauterine growth delay and 116 newborns born on time, between 38 and 41 weeks of gestation. They concluded that triglyceride concentrations were higher in the study group than in neonates born on time (45 ± 27 and 36 ± 19 mg/dL, respectively, *p* < 0.001). The findings of asymmetric growth restriction in male and female newborns revealed disparities in serum triglyceride values compared to healthy individuals. However, no differences were reported among the SGA and AGA groups concerning total cholesterol, LDL cholesterol, or HDL cholesterol [43]. In another work, Chinese researchers evaluated metabolic changes in young rats born SGA. They obtained results that corresponded to our findings, i.e., that rats born with low birth weight demonstrated significantly higher levels of total cholesterol, triglycerides, and LDL cholesterol [44]. A team of Spanish researchers, Ibanez et al., evaluated female children with precocious puberty aged 5–18 years. They noted a relationship between the occurrence of dyslipidemia and low birth weight, suggesting that detected metabolic disorders originate from the prenatal period [25]. Tenhola et al., in a Finnish population study of SGA children aged 5–12-years, found no association between body mass index (BMI) and serum lipid levels, but almost as many as half of the subjects with SGA (47.3%) were in the highest quartile for serum total cholesterol, which was adequate for children of gestational age (*p* = 0.038) [21]. Our results correspond to this study—we also did not show such differences. We obtained only a negative correlation between glucose concentration and birth length and current body weight in the SGA group. This may be due to the age of the examined children, in which the lipid profile itself does not yet clearly reflect developing metabolic disorders.

Interestingly, it was not body mass but weak dynamics of catch-up height in SGA children that increased the risk of elevated triglyceride levels at the age of 12 years almost 14 times more as compared to SGA children who had a good catch-up growth in height [21]. Krochik et al. evaluated early risk factors of metabolic syndrome occurrence in SGA children in the period prior to puberty. Although the study group showed significantly higher levels of basal insulin, basal cortisol, and uric acid, there was no statistically significant difference between the examined groups in LDL and HDL cholesterol levels [27]. Also, Umer et al. noticed that low birth weight was related to higher levels of LDL, non-HDL, and triglycerides and lower HDL levels in 11-year-old children [28]. Huang et al. studied a population of prepubertal short SGA children (aged six years). The results showed that serum concentrations of total cholesterol, triglycerides, Apo B, and Apo B/Apo A-I in this group were significantly higher. It was observed that more than 33% of short SGA children had hypercholesterolemia and 23% had hypertriglyceridemia compared to a short AGA group [45].

However, contrary to the results of this study, Evangelidou et al. [46] did not observe any significant differences in the serum total cholesterol, LDL cholesterol, triglycerides, Apo-A-1, Apo-B, and Lp(a) between an SGA group and AGA controls. The levels of Lp(a), nevertheless, were significantly higher in the SGA <3rd percentile subgroup than in the SGA 3rd-10th percentile subgroup (*p* < 0.05). In our work, we compared glucose levels between the SGA and AGA groups, demonstrating a significantly higher level in the former and a negative correlation with birth length and the current body weight.

Similar observations were made by Italian researchers [47]. Long-term studies on SGA children in two age groups (8.4 ± 1.4 years and 13.3 ± 1.8 years) evaluated lipids and glucose levels. As for glucose levels, our results corresponded to those obtained by the authors mentioned above, i.e., glucose values were noticeably higher in the two SGA groups vs. the AGA group. There were no changes found, however, in the lipid profile. It is difficult to draw conclusions about the origin of these differences. Perhaps an analysis of diet and catch-up growth, as well as ethnic differences, could help in their interpretation.

Swedish researchers conducted a prospective, longitudinal cohort study that enrolled 285 Swedish children with low birth weight and 95 children with normal birth weight (2501–4500 g). In children aged three-and-a-half and seven years, blood samples were tested and compared between the groups for the lipid profile (HDL cholesterol and LDL cholesterol) with the evaluation of apolipoproteins (ApoA1 and ApoB) as well as glucose, insulin, and homeostatic model assessment of insulin resistance (HOMA-IR). The results showed that the SGA group demonstrated a significantly higher level of mean fasting glucose than the control group. However, there were no considerable differences in levels of insulin, HOMA-IR (homeostatic model assessment for insulin resistance), or blood lipids between the SGA and AGA groups [29].

On the other hand, a study by Hong Zu Deng et al. [48] showed no differences between SGA and AGA levels of glucose and insulin, although it should be emphasized that it was conducted on a younger pediatric group.

However, attention should be paid to the limitations regarding the interpretation of our paper results. Firstly, it was a single-center study. Hence, it was carried out on a relatively small group of patients, which was its major limitation. Secondly, the study design was based on collected retrospective data, which was another limitation.

The limitations mentioned above might suggest that the derived conclusions are not readily applicable to the entire population of prepubescent SGA children.

## 5. Limitations

The main limitations of the study include recall error (part of the analysis was based on parental responses) and selection bias (only patients who agreed to participate in the study during the recruitment period were included in the research). Moreover, another limitation of this work was the relatively small group of patients and a population of only Polish children (Caucasian race). However, this was a single-center study. Future re-analysis and objective follow-up would be possible once the study group has been enlarged. 

## 6. Conclusions

This study contributes to the professional literature by demonstrating that children born SGA, with no differences in weight and BMI, have an abnormal lipid profile compared to AGA children. This could be a potential risk factor for cardiovascular diseases in adulthood. Therefore, it is justified to conduct a study on a larger group of patients born SGA. Moreover, it would be valuable to investigate in SGA patients the correlation between the results of metabolic tests and assessment of the intima-media complex, as an independent marker of cardiovascular risk. 

To prevent cardiovascular complications, this group should be closely monitored by a team of specialists, including pediatricians, cardiologists, pediatric cardiologists, endocrinologists, and pediatric nutritionists.

## Figures and Tables

**Figure 1 nutrients-15-04781-f001:**
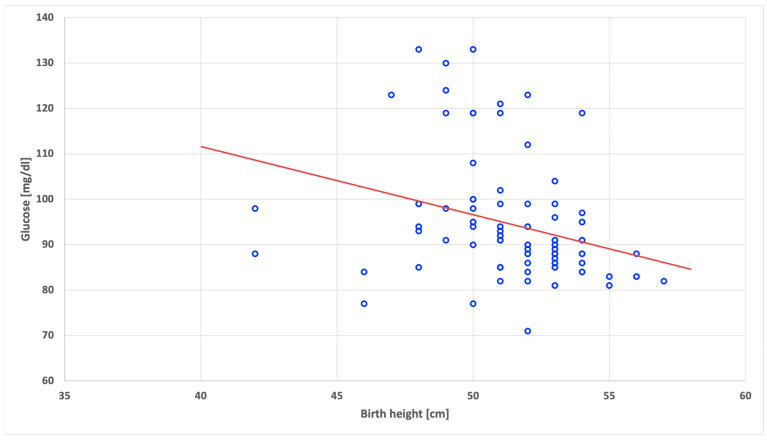
Correlation between fasting glucose level and actual birth height in the children born as small for gestational age.

**Figure 2 nutrients-15-04781-f002:**
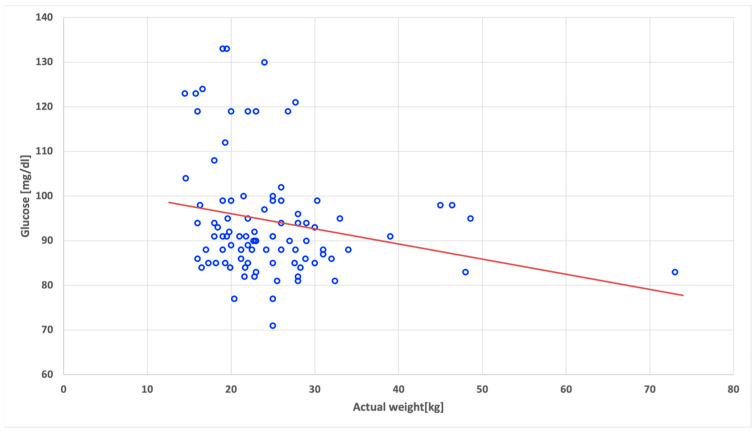
Correlation between fasting glucose level and actual body weight in the children born small for gestational age.

**Table 1 nutrients-15-04781-t001:** Characteristics of the patients.

	SGA Group (*N* = 93) Mean ± SD	Control Group (*N* = 47) Mean ± SD	*p* Value
Sex (Male/Female)	43/50	23/24	0.76 (NS)
Age on examination	7.53 ± 1.35	7.4 ± 1.77	0.67 (NS)
Birth weight (g)	2545.22 ± 209.07	3370.29 ± 503.11	*p* < 0.001
Birth height (cm)	51.43 ± 2.61	54.51 ± 3.20	*p* < 0.001
Gestational age (hbd)	38.98 ± 0.86	39.09 ± 0.74	0.52 (NS)
Actual height (cm)	123.78 ± 10.06	128.11 ± 10.18	*p* < 0.05
Actual weight (kg)	24.79 ± 8.54	26.53 ± 7.78	0.30 (NS)
BMI	15.80 ± 2.83	15.97 ± 2.43	0.76 (NS)

SGA—small for gestational age, *N*—number of children, SD—standard deviation, NS—not significant, hbd—weeks of gestation, BMI—body mass index.

**Table 2 nutrients-15-04781-t002:** Characteristics of the SGA patients.

	SGA Symmetrical (*N* = 43) Mean ± SD	SGA Asymmetrical (*N* = 50) Mean ± SD	*p* Value
Sex (Male/Female)	22/21	21/29	0.38 (NS)
Age on examination	7.72 ± 1.15	7.79 ± 1.46	0.16 (NS)
Birth weight (g)	2506.74 ± 256.66	2578.30 ± 152.28	0.51 (NS)
Birth height (cm)	49.79 ± 2.65	52.84 ± 1.53	*p* < 0.001
Gestational age (hbd)	38.94 ± 0.82	39.02 ± 0.91	0.62 (NS)
Current height (cm)	121.51 ± 9.34	125.73 ± 10.34	*p* < 0.05
Current weight (kg)	23.22 ± 7.59	26.14 ± 9.13	*p* < 0.05
BMI	15.35 ± 2.63	16.19 ± 2.97	0.12 (NS)
Ponderal index (g/cm^3^)	2.02 ± 0.23	1.75 ± 0.16	*p* < 0.00001

SGA—small for gestational age, *N*—number of children, SD—standard deviation, NS—not significant, hbd—weeks of gestation, BMI—body mass index.

**Table 3 nutrients-15-04781-t003:** Glucose and lipid profile in examined patients.

Parameter	SGA Group (*N* = 93) Mean ± SD	Control Group (*N* = 47) Mean ± SD	*p* Value
Glucose (mg/dL)	94.51 ± 12.47	84.64 ± 6.39	<0.001
Total cholesterol (mg/dL)	190.61 ± 24.66	143.23 ± 23.90	<0.001
HDL cholesterol (mg/dL)	71.46 ± 13.31	57.94 ± 18.50	<0.001
LDL cholesterol (mg/dL)	105.35 ± 20.96	82.67 ± 16.92	<0.001
Triglycerides	112.32 ± 38.10	55.89 ± 18.31	<0.001

SGA—small for gestational age, *N*—number of children, SD—standard deviation.

**Table 4 nutrients-15-04781-t004:** Glucose and lipid profile in SGA patients.

Parameter	SGA Symmetrical (*N* = 43) Mean ± SD	SGA Asymmetrical (*N* = 50) Mean ± SD	*p* Value
Glucose (mg/dL)	98.95 ± 14.79	90.56 ± 10.21	<0.001
Total cholesterol (mg/dL)	190.90 ± 27.26	188.64 ± 23.70	0.73 (NS)
HDL cholesterol (mg/dL)	70.95 ± 13.00	71.86 ± 13.76	0.85 (NS)
LDL cholesterol (mg/dL)	104.65 ± 23.71	103.03 ± 19.49	0.87 (NS)
Triglycerides	109.09 ± 38.42	111.4 ± 39.31	0.91 (NS)

SGA—small for gestational age, *N*—number of children, SD—standard deviation, NS—not significant.

**Table 5 nutrients-15-04781-t005:** Correlation coefficient between selected parameters—control group.

Parameter	Glucose	Total Cholesterol	HDL Cholesterol	LDL Cholesterol	Triglycerides
Birth weight	0.33	0.03	0.16	−0.01	0.02
Birth height	0.26	0.31	0.19	−0.11	−0.02
Actual weight	0.18	0.05	−0.24	−0.17	−0.18
BMI	0.10	0.04	−0.38 *	0.05	−0.17

* *p* < 0.05 (statistically significant).

**Table 6 nutrients-15-04781-t006:** Correlation coefficient between selected parameters—SGA group.

Parameter	Glucose	Total Cholesterol	HDL Cholesterol	LDL Cholesterol	Triglycerides
Birth weight	−0.17	−0.01	0.08	−0.13	−0.03
Birth height	−0.40 *	0.09	0.09	0.13	−0.04
Actual weight	−0.22 *	−0.01	−0.002	−0.03	0.08
BMI	−0.16	−0.09	−0.06	−0.08	0.10

* *p* < 0.05 (statistically significant).

**Table 7 nutrients-15-04781-t007:** Correlation coefficient between selected parameters—SGA symmetrical group.

Parameter	Glucose	Total Cholesterol	HDL Cholesterol	LDL Cholesterol	Triglycerides
Birth weight	−0.14	−0.14	0.09	−0.07	−0.11
Birth height	−0.18	0.27	0.45 *	0.23	−0.002
Actual weight	−0.26	−0.05	−0.05	−0.14	0.01
BMI	−0.15	−0.18	−0.19	−0.22	0.08

* *p* < 0.05 (statistically significant).

**Table 8 nutrients-15-04781-t008:** Correlation coefficient between selected parameters—SGA asymmetrical group.

Parameter	Glucose	Total Cholesterol	HDL Cholesterol	LDL Cholesterol	Triglycerides
Birth weight	−0.06	0.03	0.01	0.07	0.001
Birth height	−0.27 *	0.10	−0.06	0.16	−0.04
Actual weight	−0.08	0.11	−0.01	0.09	0.11
BMI	0.03	−0.05	−0.04	−0.04	−0.006

* *p* < 0.05 (statistically significant).

## Data Availability

Data are contained within the article.

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
