# Peer review of "Lipid Profile in Children Born Small for Gestational Age"

_nutrients, 2023, doi:10.3390/nu15224781_

Round 1

Reviewer 1 Report (Previous Reviewer 1)

Comments and Suggestions for Authors

Zamojska et al. Present very interesting data regarding an unfavorable lipid profile in children that had been born small for gestational age. Though the topic is very interesting and the study is well conducted, there are some issues to be addressed:

Introduction/Discussion: The authors need to clarify in the introduction what the research gap is they´d like to fill. Since there are many previous studies in the field, it is necessary to discuss the novelty of the presented data.  In this respect it would be important to better discriminate between studies investigating low birth weight, premature birth, and those investigating SGA.

The method sections lacks a description of the performed statistics.

Figure 1 / lines 182ff: Since the values do not look normally distributed, Spearman correlation analysis would be correct. Was this performed?

Results section: I think it would be advantageous to present all those correlations in a table.

Table 1: Birth height would be interesting to be included.

Is it possible to quantitate catch-up growth and to include it into the analyses?

Author Response

Thank You very much for your valuable remark and all suggestions. 
1. We supplemented the introduction with data on low birth weight, prematurity, and children born small for gestational age. We gave the definition of LBW and SGA and highlight the difference in the definition of LBW, SGA, IUGR. 
   Our study group focused on SGA patients born at term.  In the discussion, we referred to the work carried out on term-SGA patients. We provide information why it is useful to test this group of patients.
2. We made corrections and supplemented the text with missing elements of statistical analysis.
3. As you suggested we aded all correlations in tables (table 5,6,7,8).
4. We added birth height to the table 1.
5. Thank you very much for this suggestion. Our study was designed as a retrospective analysis. 
   Due to the retrospective nature of the staudy, unfortunately, we did not have opportunity to obtain information about specific weight gains in the studied patients.
   Therefore, we could not comment on catch-up growth in our research. 

Reviewer 2 Report (New Reviewer)

Comments and Suggestions for Authors

The authors reported change in lipid profiles in SGA born children.

I strongly agree to the authors idea that says distinguishing SGA from LBW is important because its etiology is different.

 I have several requests:

Introduction

Authors describe previous studies but I recommend to emphasize the importance of distinguishing SGA from LBW.

 Material and methods

Are control participants matched with gestational age? If not, please add information about gestational age information to characteristic table.

 Please clarify how the authors defined symmetrical and asymmetrical IUGR.

Line 120- 123 should be moved to result part, or describe that participants with bad condition were excluded, but none of them was in bad condition.

Result

Authors analyze correlations using birth weight but can you analyze using birth weight z-score? Because birth weight differs by gestational age.

Line 282: Shape of bracket is different

Authors discuss about sex in discussion part but they do not consider in their analysis. Please mention in limitation part, or if they analyze with birth weight z-score, sex information is included, too so it will be fine.

I look forward to hear from you, Thank you.

Author Response

Thank You very much for your valuable remark and all suggestions. 
1. We have updated the introduction according to your suggestions - We gave the definition of LBW and SGA and highlight the difference in the definition of LBW, SGA, IUGR.
2. Control participants matched with gestational age to studied group - this information is in table 1 and in the text in section Results: "However, there was no significant difference between the groups regarding gestational age."
3. We have explaind in text how we diveded patients into symmetrical and asymmetrical IUGR. We also add PI values into table 2.  
4. We have moved line 120-123 to results part as you suggested. 
5. Thank you for your suggestion. We did not use the birth weight z-score to estimate the correlation because there was no statistically significant difference in gestational age in the analyzed group and all children in the study were born at term.
6. Thank You for noticing different shape of bracket - we have corrected this. 
7. We added information about sex of examined patients (we added this in characteristic tabels and additional correlations as well).

Reviewer 3 Report (New Reviewer)

Comments and Suggestions for Authors

In the manuscript submitted to me for review entitled "Lipid profile in children born small for gestational agethe authors Justyna Zamojska, Katarzyna Niewiadomska-Jarosik, Beata Kierzkowska, Marta Gruca, Agnieszka Wosiak, Elżbieta Smolewska assessed the lipid profile in early childhood in the Polish children population born small for gestational age (SGA). The study was conducted with 93 children (48 girls and 45 boys) aged 5-11 years.

The topic of the research is extremely interesting, given that the number of babies born with a lower than normal weight for their gestational age is increasing due to the stressful lifestyle and the increasingly frequent health problems of pregnant women. It is important to monitor the health status of such children in order to avoid possible health problems over the years.

The authors' study covers scientific data published over the past nearly six decades. To support their research, the authors use 43 references, of which 1/4 are from the last 5 years. The results are clearly presented using 2 tables and 1 figure.

My remarks and recommendations to the authors are:

1.     I am interested in whether there have been studies conducted with children with lower birth weight compared to normal for gestational age and any data on the way they are fed. Is there a possibility that a change in lipid profile could be due to the different food, such as type and amount, that they ate especially in the first two years (but not necessarily for that exact time period). However, there is also the possibility that in some children the parents factor also exists. Parents' concern that the child was born underweight could lead to a drive to catch up, which could even lead to a change in the lipid profile at a later stage.

2.     2. I had the expectation that since the number of girls and boys included in the study is noted at the beginning of the methods, that when presenting the results in the tables in separate columns, the results for both sexes would be indicated, which I think would have enriched the results.

3. On line 241 "since991" is written. I guess there is some misprint in writing.

4. Some of the references do not list all the authors (Nos. 5, 6, 7, 14, 15, 17, 20, 21, 22, 23, 24, 27, 28, 29, 30, 32, 33, 36, 38, 39, 40, 41, 42 and 43). Personally, when I read an article, I prefer the references to be fully written, instead of having to search for some of the authors. I think it would be helpful to your readers if all authors in all references are listed.

Author Response

Thank You very much for your valuable remark and all suggestions. 
1. Unfortunately, we did not have information on feeding children. This is a very valuable comment, as well as regarding the phenomenon of catch-up growth. Due to the retrospective nature of our study, we did not have the opportunity to obtain and analyze certain information.
2. According to your suggestion we added results concerning gender of patients.
3. We correct refferences as you suggested. 

Reviewer 4 Report (New Reviewer)

Comments and Suggestions for Authors

The research topic is of significant interest, contributing to the field of pediatric health and the understanding of long-term outcomes associated with birth weight and gestational age. The study's design, methodology, and analysis provide a valuable addition to existing literature. However, there are several areas where improvements could be made to enhance the clarity, rigor, and impact of the findings.

Expand on Literature Review: The introduction could benefit from a more comprehensive review of previous studies, particularly focusing on the long-term metabolic impacts of low birth weight. This would help to position your study more firmly within the existing body of research.

Methodological Detailing: Provide more details about the study's methodology, including sample size determination, recruitment strategies, and the specifics of the statistical analysis. This information is crucial for assessing the study's validity and reproducibility.

Results Interpretation: While the results are presented clearly, there is room for a more in-depth discussion on how these findings compare with existing studies. Any unexpected or novel results should be highlighted and explored further. Review the significant findings and provide a direct comparison with existing studies, particularly those referenced in the manuscript. Highlight whether the findings align or diverge from established literature and discuss possible reasons for any discrepancies.

Discussion of Limitations: Elaborate on the limitations of your study, such as potential confounding factors, the generalizability of the findings, and any biases that could have affected the results. Discuss how these limitations might impact the interpretation and application of your findings. Specify the limitations related to the sample size and demographic constraints, which may affect the generalizability of the study's outcomes.

Future Research Directions: Conclude with suggestions for future research, including how your findings can be built upon, and what new questions or hypotheses your study raises. Suggest potential avenues for future research, such as longitudinal studies to track the long-term cardiometabolic health of the subjects or expanding the cohort to include a more diverse population. This would not only give your paper a forward-looking perspective but also help in identifying areas that need further exploration.

Comments on the Quality of English Language

The manuscript is generally well-written, with clear and concise language. However, there are a few areas where improvements could be made to enhance readability and comprehension:

Consistency in Terminology: Ensure consistency in the use of technical terms and abbreviations throughout the manuscript. This will help maintain clarity and avoid confusion for readers.

Sentence Structure: In some sections, the sentence structure is complex or awkward, which might hinder understanding. Simplifying these sentences could improve the overall readability.

Grammar and Punctuation: Minor grammatical and punctuation errors are present, which should be corrected to maintain the manuscript's professional quality.

Proofreading and Editing: A thorough proofreading and editing pass is recommended to catch and correct these issues. Consider seeking a professional language editing service if necessary, to ensure the manuscript meets the publication standards.

Author Response

Thank you very much for your valuable remark.
1. According to Your suggestions we expanded on literature on long-term metabolic impacts of low birth weight.  
2. Thank you very much for your valuable remark. Our study was designed as a retrospective analysis, which inherently involved examining existing data collected for clinical, not research, purposes. Due to the retrospective nature of the study, we did not have the opportunity to influence the size of the compared groups. 
The sample size was thus determined by the number of available records that met our inclusion criteria during the specified time frame of our study.
As for the recruitment strategies, given the retrospective design, our 'recruitment' was based on a comprehensive search of our institution's medical records.
We acknowledge that the retrospective design may introduce certain limitations. However, we believe that the study's findings are valuable in contributing to the existing literature, providing meaningful insights and identifying areas for future prospective research.
3. We have significantly expanded the results with tables regarding the examined correlations. We also included the results of the SGA analysis by gender and obtained statistically significant correlations. 
We have also tried to refer to the obtained results in the discussion, expanding it with additional fragments.
4. As you suggested we have added limitations of our study and conclusion section with future research direction. 
5. According to Your suggestions, the manuscript was linguistically proofread at the Language Center of the Medical University of Lodz, Poland. 

Round 2

Reviewer 4 Report (New Reviewer)

Comments and Suggestions for Authors

The authors arranged the paper well following my instructions

Comments on the Quality of English Language

English has been improved 

This manuscript is a resubmission of an earlier submission. The following is a list of the peer review reports and author responses from that submission.

Round 1

Reviewer 1 Report

Comments and Suggestions for Authors

Zamojska et al. Present very interesting data regarding an unfavorable lipid profile in children that had been born small for gestational age. Though the topic is very interesting and the study overall is well conducted, there are some issues to be addressed:

Introduction/Discussion: The authors need to clarify in the introduction what the research gap is they´d like to fill. Since there are many previous studies in the field, it is necessary to discuss the novelty of the presented data.  In this respect it would be good to better discriminate between studies investigating low birth weight, premature birth, and those investigating SGA. Also the discussion could be more concise.

The method sections lacks a description of the performed statistics.

Figure 1 / lines 182ff: Since the values do not look normally distributed, Spearman correlation analysis would be correct. Was this performed?

All results of correlations with low r and very high p values should be named “no correlation” rather than “positive or negative correlation with no significant meaning”.

Results section: I think it would be advantageous to present all those correlations in a table.

Table 1: Birth height would be interesting to be included.

Is it possible to quantitate catch-up growth and to include it into the analyses?